# Tissue-Specific Human Extracellular Matrix Scaffolds Promote Pancreatic Tumour Progression and Chemotherapy Resistance

**DOI:** 10.3390/cells11223652

**Published:** 2022-11-17

**Authors:** Walid Al-Akkad, Pilar Acedo, Maria-Giovanna Vilia, Luca Frenguelli, Alexander Ney, Irene Rodriguez-Hernandez, Peter L. Labib, Domenico Tamburrino, Gabriele Spoletini, Andrew R. Hall, Simone Canestrari, Anna Osnato, Jose Garcia-Bernardo, Leinal Sejour, Vessela Vassileva, Ioannis S. Vlachos, Giuseppe Fusai, Tu Vinh Luong, Steven R. Whittaker, Stephen P. Pereira, Ludovic Vallier, Massimo Pinzani, Krista Rombouts, Giuseppe Mazza

**Affiliations:** 1UCL Institute for Liver and Digestive Health, Royal Free Hospital, University College London, London NW3 2PF, UK; 2Engitix Therapeutics, The Westworks, 195 Wood Lane, Shepherd’s Bush, London W12 7FQ, UK; 3Division of Surgery, Royal Free London NHS Foundation Trust, University College London, London NW3 2QG, UK; 4Sheila Sherlock Liver Centre, Royal Free London NHS Foundation Trust, London NW3 2PF, UK; 5Wellcome Trust-MRC Cambridge Stem Cell Institute, Anne McLaren Laboratory, University of Cambridge, Cambridge CB2 0AW, UK; 6Wellcome Trust Sanger Institute, Hinxton, Cambridgeshire CB10 1RQ, UK; 7Cancer Research Institute, HMS Initiative for RNA Medicine, Department of Pathology, Beth Israel Deaconess Medical Center, Harvard Medical School, Boston, MA 02115, USA; 8Broad Institute of MIT and Harvard, Cambridge, MA 02142, USA; 9Department of Surgery and Cancer, Imperial College London, London SW7 2AZ, UK

**Keywords:** tissue engineering, 3D cell-culture, tissue-specificity, extracellular matrix, pancreatic ductal adenocarcinoma, liver metastasis, chemoresistance

## Abstract

Over 80% of patients with pancreatic ductal adenocarcinoma (PDAC) are diagnosed at a late stage and are locally advanced or with concurrent metastases. The aggressive phenotype and relative chemo- and radiotherapeutic resistance of PDAC is thought to be mediated largely by its prominent stroma, which is supported by an extracellular matrix (ECM). Therefore, we investigated the impact of tissue-matched human ECM in driving PDAC and the role of the ECM in promoting chemotherapy resistance. Decellularized human pancreata and livers were recellularized with PANC-1 and MIA PaCa-2 (PDAC cell lines), as well as PK-1 cells (liver-derived metastatic PDAC cell line). PANC-1 cells migrated into the pancreatic scaffolds, MIA PaCa-2 cells were able to migrate into both scaffolds, whereas PK-1 cells were able to migrate into the liver scaffolds only. These differences were supported by significant deregulations in gene and protein expression between the pancreas scaffolds, liver scaffolds, and 2D culture. Moreover, these cell lines were significantly more resistant to gemcitabine and doxorubicin chemotherapy treatments in the 3D models compared to 2D cultures, even after confirmed uptake by confocal microscopy. These results suggest that tissue-specific ECM provides the preserved native cues for primary and metastatic PDAC cells necessary for a more reliable in vitro cell culture.

## 1. Introduction

Pancreatic ductal adenocarcinoma (PDAC) accounts for over 90% of all pancreatic cancers [1], with 450,000 new cases diagnosed globally per year [2]. PDAC is currently the third leading cause of cancer-related death in the United States [3] and is projected to become the second by 2030 [4]. The striking numerical similarity between incidence and mortality rates highlight the dismal prognosis of this disease. The median overall survival is less than six months, and the 5-year worldwide survival rate is less than 8% [5,6,7]. Indeed, this disease is usually incurable at diagnosis as patients are commonly diagnosed at a metastatic or locally advanced stage, which is mostly due to PDAC’s almost asymptomatic clinical course [8]. There is an evident lack of translating good clinical response into relative survival benefits, with median overall survival (OS) of 8.1 months and progression-free survival ranging from 2.4 to 11.0 months [9]. Underlying reasons for therapeutic failures include biological and physiological chemo-resistance, which are thought to be mediated largely by intrinsic cellular mutations in neoplastic cells and a reactive stroma. This mostly composed of cancer-associated fibroblasts (CAF), activated pancreatic stellate cells (PSC), and the extracellular matrix (ECM) [10], in the context of an evident desmoplastic reaction [11,12,13,14,15,16,17]. The desmoplastic reaction exerts both bio-mechanical and biochemical effects on tumour and stromal cells [18]. More importantly, common PDAC metastatic sites (e.g., liver) show pathological resemblance to the primary tumour with analogous ECM components [16].

We propose the use of 3D ECM scaffolds that could redefine in vitro models of PDAC and enable effective preclinical testing of novel therapies. Initially, the elimination of native cellular material, the preservation of ECM proteins and the micro-architecture of decellularized human pancreata and liver was confirmed. Cell seeding with both primary (PANC-1, MIA PACA-2) and metastatic (PK-1) pancreatic cancer cells demonstrated a conserved tissue-specific invasive behavior depending on the 3D ECM structure of origin. Moreover, we observed a striking alteration in cell response to different cancer therapies in the presence of a natural ECM niche, which was further investigated with next-generation RNA sequencing (RNAseq). These findings offer new insights into the understanding of the acellular microenvironment in PDAC progression and metastasis, and they support further application of this platform in target/biomarker discovery as well as drug screening.

## 2. Materials and Methods

### 2.1. Tissue Retrieval and Preparation

The study was approved by the UCL Royal Free Biobank Ethical Review Committee (NRES Rec Reference: 11/WA/0077). Informed consent for the research was confirmed via the NHSBT ODT organ retrieval pathway, and the project was also approved by the NHSBT Research Governance Committee. Donor organs were processed in accordance with the UCL Royal Free Biobank protocols under the Research Tissue Bank Human Tissue Act license, prior to use for research. Healthy human livers and pancreata not suitable for transplantation were obtained from the Royal Free BioBank. Pancreata and livers were washed and perfused with 1% phosphate buffered saline (PBS; Sigma Aldrich, St. Louis, MO, USA) to clear any residual blood, then air-dried for 10 min and frozen at −80 °C. 

### 2.2. Liver Cube Decellularisation

Human liver cubes were initially thawed in a water bath at 37 °C for 1 hour (h), followed by the addition of 1.2 mL of 1% PBS for 15 min. Once thawed, the cubes were transferred to 2 mL safe-lock tubes (Eppendorf, Stevenage, UK). A standardized 1.5 mL of each solution was added to its respective tube and agitated at 40 RCF for each step. The regime for the decellularization of the liver cubes is shown in Appendix A and is based on Mazza et al. [19]. The reagent mixture solution was prepared as follows:

Reagent Mixture Solution: 3% sodium deoxycholate, BioXtra, ≥98.0 (Sigma-Aldrich), 0.5% sodium dodecyl sulfate, BioXtra, ≥99.0 (Sigma-Aldrich), 0.3% Triton X-100 (Sigma-Aldrich), 0.0025% Gibco^®^ trypsin-EDTA (Life Technologies, Carlsbad, CA, USA) and 4.3% of sodium chloride (Sigma-Aldrich) in deionized water (MilliQ by Millipore, Burlington, MA, USA) and stirred for 1 h using a magnetic stirrer.

### 2.3. Pancreas Decellularisation by Perfusion

Whole human pancreata were partially thawed at 4 °C for 16 h. The spleen and peripancreatic tissue were carefully removed whilst maintaining the integrity of the pancreatic capsule. The splenic and mesenteric arteries and the superior mesenteric vein were ligated. Excess duodenal–jejunal tissue was removed and the proximal duodenum was stapled closed. Finally, 1% PBS was pumped into the pancreas through the portal vein and any leaks from the pancreas–duodenum block were ligated. Next, the pancreata were cannulated through the portal vein and were decellularized using the Harvard Apparatus ORCA bioreactor (*n* > 3). The bioreactor used for the decellularization was set up as shown in Appendix A. The software used to control and monitor the perfusion system was HART v1.0.0.0 (Harvard Apparatus, Holliston, MA, USA). The perfusion regime for the decellularization is shown in Appendix A and is loosely based on previously published work [20]. After decellularization was complete, the pancreata were dissected into ~5 × 5 × 5 mm cubes and stored in either: 1X HBSS (Thermofisher Scientific, Waltham, MA, USA) at 4 °C until future experiments or fixed in 4% formaldehyde and assessed by histology and immunohistochemistry. The PAA solution was prepared by adding 0.1% peracetic acid and 4% absolute ethanol in distilled water.

### 2.4. DNA Quantification

Fresh and decellularized tissue samples marked for DNA quantification were retrieved from the −80 °C freezer and thawed in a 37 °C water bath for 1 h. The cubes were then weighed and if necessary, cut to be between 15 and 25 mg in mass. The cubes were then placed in 1.5 mL microcentrifuge tubes. Twenty μL of proteinase K was added to each one, and then mixed thoroughly using a vortex. The cubes were then placed into a heating block at 56 °C for at least 16 h or until they were completely lysed. The DNA was then extracted using the QIAGEN DNAeasy Blood and Tissue Kit according to the manufacturer’s instructions. The extracted DNA was eluted in 200 µL of buffer AE and was quantified using a NanoDrop ND-2000 spectrophotometer.

### 2.5. 2D Cell Culture

Three cell lines were used for this study: PANC-1 (ATCC, Manassas, VA, USA), MIA PACA-2 (ATCC) and PK-1 (kindly provided by Professor Stephen Pereira). All the cells were cultured in RPMI 1640 medium (Thermofisher Scientific) supplemented with 2 mM/L glutamine (Thermofisher Scientific), 10% Foetal Bovine Serum (FBS; Thermofisher Scientific) and 1% 1X Antibiotic-Antimycotic (Thermofisher Scientific). All the cells were cultured under standard conditions in a humidified, 5% CO_2_ and 37 °C incubator. Once the cells reached ~75% confluence, they were split at a ratio of 1:3 using 0.25% Trypsin-EDTA (Thermofisher Scientific).

### 2.6. Scaffold Sterilization and Preparation for Bioengineering

To prepare the decellularized tissue for cell culture, scaffolds were sterilized using 1.5 mL of PAA solution for 45 min in an orbital shaker (Labnet-Orbit™ M60 microtube shaker) at 700 rpm. This was followed by replacing the solution with sterile 1X HBSS (Thermofisher Scientific) for 15 min in the orbital shaker. The sterile scaffolds were then placed in individual wells in a 48-well plate 24 h prior to the additions of cells and 1.4 mL of RPMI 1640 medium supplemented with 2 mM/L glutamine, 10% FBS and 1% 1X Gibco^®^ antibiotic–antimycotic was added. 

### 2.7. Three-Dimensional Cell Cultures

Scaffolds were kept overnight in media as mentioned above (day c1). Just prior to cell seeding, the scaffolds were transferred to individual wells in a 96-well plate. Cells (PANC-1, MIA PaCa-2 and PK-1) were resuspended at a density of 0.5 million cells per 20 µL per scaffold (*n* ≥ 4 scaffolds per cell line). Cells were drawn up using a pipette and released on top of the decellularized tissue. Seeded scaffolds were kept for 2 h in a humidified incubator at 37 °C with 5% CO_2_ allowing cell attachment. This was followed by the addition of 120 µL of culture medium and left in a humidified incubator at 37 °C with 5% CO_2_ overnight. The following day, scaffolds were transferred to individual wells in a 48-well plate and 1 mL of culture medium was added. Thereafter, wells and fresh media were changed every 3 days. At days 7 and 14 following seeding, the scaffolds were placed in either 4% formaldehyde and assessed by histology and immunohistochemistry or snap-frozen and stored for further gene expression analysis. 

### 2.8. Sample Processing for Histology and Immunohistochemistry

Fresh, decellularized and bioengineered tissue samples, previously fixed in 4% formaldehyde, were retrieved, dehydrated in a series of industrial IDA (Acquascience, Richmond, RI, USA) and xylene (Acquascience) baths and finally embedded in paraffin. The samples were sliced into 4 µM sections using a Leica RM2035 microtome (Leica biosystems, Danvers, MA, USA). All the sections were then passed through three histology grade xylene baths for a minimum of 5 min, and then through three IDA baths for a minimum of 2 min, finally ending up in tap water.

### 2.9. Histology Staining

The sections were stained at room temperature as follows:

Haematoxylin and Eosin (H&E): The sections were treated with haematoxylin Harris’ formula (Leica biosystems) for 10 min and then washed in tap water for 5 min. Next, the sections were stained with eosin (Leica biosystems) for 3 min, and then washed again with water. The sections were then dehydrated through IDA and placed in histology grade xylene until mounted.

Picro-Sirius Red (SR): the sections were treated with freshly filtered Picro-Sirius red–F38 (R.A. Lamb; CI-35780; Epredia, Kalamazoo, MI, USA) for 20 min. The sections were then dehydrated with IDA and placed in histology-grade xylene until mounted.

### 2.10. Immunohistochemistry

Manual IHC staining: the slides were incubated in 0.5% trypsin (MP Biomedical)/0.5% chymotrypsin (Sigma-Aldrich)/1% calcium chloride (BDH) in 10% Tris-buffered saline (TBS) for 30 min at 37 °C. The slides were then washed in 10% TBS at pH 7.6 with 0.04% Tween-20 (Sigma-Aldrich) for 5 min. The slides were later blocked in peroxide blocking solution (Novocastra) for 5 min and incubated for 1 h in the following primary antibodies; collagen I (Rabbit pAb to coll1 (ab34710), diluted 1:200; Abcam, Cambridge, UK), collagen III (Rabbit pAb to coll3 (ab7778), diluted 1:500; Abcam), collagen IV (mouse mAb to coll4 (M0785), diluted 1:25; Dako) and fibronectin (mouse mAb to fibronectin (MAB1937), diluted 1:100; Millipore). The slides were then placed for 25 min in Novolink^TM^ post primary (Novocastra), 25 min in Novolink^TM^ polymer solution (Novocastra) and developed with Novolink^TM^ 3,3′ di-amino-benzidine (Novocastra). The slides were finally counterstained with Mayer’s haematoxylin (Sigma-Aldrich) for 1 min. All the sections were mounted with DPX (Leica biosystems) and cover-slipped.

Automated IHC staining: this was performed on the Leica Bond III Automated Immunostaining platform, using Leica Bond Polymer Refine detection with a DAB chromogen (Leica, DS9800). γH2A (Cell Signaling, monoclonal Ser139, #9718; Danvers, MA, USA) is diluted 1/480 and applied at room temperature for 40 min, following on-board HIER using Leica Epitope Retrieval 1 (ER1) solution for 30 min (low pH). ASP175 (Cell Signaling, polyclonal D175, #9661) was diluted 1/300 and applied at room temperature for 40 min, following on-board HIER using Leica Epitope Retrieval 1 (ER1) solution for 30 min (low pH).

All the slides were observed using a Zeiss Axioskop 40. Images were captured with an Axiocam IcC5 using Zeiss Axiovision (version 4.8.2, Zeiss, Oberkochen, Germany). All the images were analyzed and enhanced using Fiji (ImageJ, version 1.49d, National Institutes of Health, Bethesda, MD, USA).

### 2.11. Chemotherapy Treatments

To determine the most suitable concentration of gemcitabine and doxorubicin for use in 3D experiments, 5 × 10^3^ cells of PANC-1, 7.5 × 10^3^ cells of MIA PaCa-2 and 7.5 × 10^3^ cells of PK-1 cells were seeded on individual wells in a 96-well-plate (*n* = 6 per condition). Two hundred µL of culture media was added to each well and the cells were incubated for 24 h in a humidified incubator at 37 °C with 5% CO_2_. Next, 200 µL of 9 different concentrations of gemcitabine (0.01, 0.02, 0.04, 0.08, 0.1, 0.2, 0.3, 0.4 and 0.5 µM) and doxorubicin (0.001, 0.005, 0.01, 0.02, 0.05, 0.1, 0.2, 0.5 and 0.8 µM), as well as a negative control of culture medium and a positive control of 10% DMSO in culture media, were added to their appropriate wells. The cells were treated with the above-mentioned concentrations for 24 h. Finally, all the solutions were discarded and 200 µL of fresh culture medium was added to each well. Cells were left for 96 h in a humidified incubator at 37 °C with 5% CO_2_, to assess cell viability post-treatment using the Alamar Blue assay.

The drugs were prepared as follows:

Gemcitabine: a stock solution of 1 mM was prepared by solubilizing 0.026% (*w*/*v*) of gemcitabine powder (Gemzar, Eli Lilly and Company, Indianapolis, IN, USA) in distilled water. To obtain the desired drug concentrations, the stock solution was diluted in culture media.

Doxorubicin: a stock solution of 1 mM was prepared by solubilizing 0.058% (*w*/*v*) of doxorubicin hydrochloride powder (Sigma-Aldrich) in DMSO (Sigma-Aldrich). To obtain the desired drug concentrations, the stock solution was diluted in culture media.

### 2.12. Treatment of 3D Bioengineered Scaffolds with Chemotherapeutics

Both liver and pancreas scaffolds were prepared and seeded with PANC-1, MIA PaCa-2 or PK-1 cells as described above and cultured for 9 days. On the 9th day the media was discarded and 1.4 mL of 0.5 μM gemcitabine or 0.5 μM doxorubicin was added to the appropriate scaffolds. As a negative control, 1.4 mL of media was added to their respective scaffolds. The scaffolds were then allowed to incubate in the dark for 24 h in a humidified incubator at 37 °C with 5% CO_2_. Next, the scaffolds were moved to fresh wells and washed with 1.4 mL 1X HBSS for 5 min and 1.4 mL of fresh media was added to each scaffold. The scaffolds were then left for 96 h in a humidified incubator at 37 °C with 5% CO_2_, to assess cell viability post-treatment using the Alamar Blue assay.

### 2.13. Cell Viability Assay

For experiments performed on 2D plastic, the culture media was discarded, and the cells were washed three times with 200 µL 1X HBSS. Residual HBSS was discarded and 200 µL of a 10% Alamar Blue solution (Thermofisher Scientific) in culture media was added to each well. The cells were allowed to incubate with the Alamar Blue in the dark for 2.5 h in a humidified incubator at 37 °C with 5% CO_2_.

For experiments performed in the 3D scaffolds, culture media was discarded, and the scaffolds were washed three times with 1.4 mL 1X HBSS. Residual HBSS was discarded and 1.4 mL of a 10% Alamar Blue solution in culture media was added to each well. The scaffolds were allowed to incubate with the Alamar Blue in the dark for 2.5 h in a humidified incubator at 37 °C with 5% CO_2_.

Fluorescence was measured immediately after incubation on a FLUOstar Omega fluorescence microplate reader (BMG Labtech, Ortenberg, Germany) and quantified using excitation and emission wavelengths of 540 nm and 595 nm, respectively. The data measured in arbitrary units for the treated samples were normalized to the negative control (non-treated samples) and reduction in percent (%) survival was calculated.

### 2.14. Confirmation of Doxorubicin Uptake 

To confirm the uptake of the chemotherapeutic agent by the cells in the 3D scaffolds, doxorubicin fluorescence was used [21]. PANC-1 and MIA PaCa-2 cells on pancreas scaffolds and PK-1 cells on liver scaffolds were cultured for 9 days as described above. On day 9, the media was discarded and 1.4 mL of a 0.5 μM doxorubicin solution in cell culture media or fresh media was added to 3 samples of each condition followed by incubation in the dark for 24 h in a humidified incubator at 37 °C with 5% CO_2_. Next, the scaffolds were washed with 1X PBS for 5 min, transferred to a mold with OCT (Agar Scientific, Stansted, UK) and snap-frozen in liquid nitrogen. Twenty micrometer sections were cut using a CRYOTOME FSE cryostat (Thermofisher Scientific). All sections were then washed 3 times for 5 min with 1X PBS and stained with 300 nM DAPI (Thermofisher Scientific) for 1 min. The slides were imaged with a BX63 fluorescence microscope (Olympus, Tokyo, Japan) using excitation/emission of 350/460 nm (DAPI), 495/525 nm (collagen) and 545/620 nm (doxorubicin). The images were then processed with the software Fiji (ImageJ, version 1.52i, National Institutes of Health, Bethesda, MD, USA).

### 2.15. RNA Extraction 

Total RNA was extracted from 3D cultures using TRIzol reagent (Qiagen, Hilden, Germany) and RNeasy Universal Mini Kit (Qiagen) as described by the manufacturer’s instructions. Briefly, the 3D frozen samples were left to incubate at room temperature with 650 µL of TRIzol in a 2 mL safe-lock Eppendorf tube for 20 min and was followed by the addition of 7 mm stainless steel bead. The content was then agitated at 30 Hz for 8 min in a TissueLyser II (Qiagen). The content of the tube (excluding the bead) was then transferred to a new 1.5 mL Eppendorf tube and the manufacturer’s protocol was followed from step 4. 

### 2.16. Next Generation RNAseq

RNA extracted as described above was used for library preparation and sequencing at the Wellcome Sanger Institute next-generation sequencing facility (Cambridge, UK). A PolyA purified opposing strand library kit was used, and 4 samples per lane were multiplexed in 6 lanes on Illumina HiSeq 2000, 2 × 75 bp, Paired-End reads. Seventy-five to one hundred and twenty million pairs were sequenced per sample. 

### 2.17. RNAseq Data Analysis

The read quality was evaluated using FastQC and the data were pre-processed with Cutadapt [22] for adapter removal following the best practices. Gene expression quantification was performed using Salmon [23] with GC and fragment length bias correction against GRCh38 human transcriptome derived from Ensembl 98 [24]. Raw counts were normalized using the method of trimmed mean of M-values (TMM) [25] using the calcNormFactors function in edgeR. The normalization factors calculated here were used as a scaling factor for the library sizes. The voom transformation was applied to the normalized data, which were finally used in linear models and empirical Bayes statistics using limma R package for differential gene expression analyses. False discovery rates were estimated using fdrtool. ClusterProfiler [26] was utilized for functional enrichment investigations using Hallmark gene sets, C2 Canonical pathways gene sets and C5 Gene Ontology Biological Processes gene sets from the Molecular Signatures Database (MSigDB). All these analyses and visualizations were performed using R/Bioconductor. STRING analysis was used to investigate potential protein-protein interaction networks based on the differential gene expression data [27]. Gene Set Enrichment Analysis (GSEA) [28] was performed using GSEA software v4.3.0 (http://www.broadinstitute.org/gsea/index.jsp, Cambridge, MA, USA; accessed on 22 February 2021) with the specific settings: permutations-1000, permutation type-gene set, metric for ranking genes-t-test. All the RNAseq data are available in the SRA repository under accession number PRJNA890533.

### 2.18. Statistics and Data Analysis

A minimum of three biological replicates were performed for all the protocols. Statistical significance was determined using: (i) an ordinary one-way ANOVA followed by a Tukey’s multiple comparison test for DNA quantification and qRT-PCR, (ii) a two-way ANOVA followed by a Tukey’s multiple comparison test for cell viability analysis and (iii) a Kruskal–Wallis one-way ANOVA followed by a Dunn’s multiple comparison test for cell-size analysis. All statistical analyses and graphs were generated using GraphPad Prism (version 8.1.0, Dotmatics, San Diego, CA, USA).

## 3. Results

### 3.1. Characterisation of the Decellularized Human Pancreata

Decellularisation of the en bloc pancreata was achieved within two weeks of perfusion. The protocol for decellularization was loosely based on an established protocol used for healthy human liver decellularization [20]. During and following decellularization, the pancreas gradually turned from its natural brown-pink color (Figure 1a) to translucent white with the dissolution of cells (Figure 1b). The decellularization protocol, based on a retrograde perfusion through the portal vein, included a combination of four different cell elimination methods: (i) physical cell-damaging by freezing and thawing, (ii) osmotic shock to allow cell lysis, (iii) detergents to destruct chemical bonds, and (iv) flow shear stress to allow the penetration into the parenchyma of the pancreas to eliminate cellular remnants. The elimination of cellular material was histologically evaluated by Picro-Sirius Red (SR) (Figure 1c,d,k,l) and H&E staining (Figure 1g,h,m,n), which showed no signs of cellular content or nuclear material, respectively. DNA quantification supported the histological analysis and the measured DNA content was less than 120 ng/mg-wet tissue in all three segments of the pancreas (Appendix A). This was found to be significantly lower (>10-fold) than the native pancreas tissue (*p* < 0.0001). Next, the head, body, and tail of the decellularized pancreata were investigated histologically to verify the preservation of three hallmark features of the native tissue: (i) islets of Langerhans (Figure 1k; black arrow), (ii) exocrine ECM ducts (Figure 1k; green arrow) and (iii) blood vessels (Figure 1k; blue arrow) were observed. 

To further investigate the retention of ECM proteins (namely collagen I, collagen III, fibronectin, and laminin), immunohistochemistry was performed on a fresh (Figure 1g–j) and decellularized (Figure 1n–r) pancreas. All the stains showed similar patterns to the native pancreas tissue. Collagen I (Figure 1o), collagen III (Figure 1p) and fibronectin (Figure 1q) stained positive for the exocrine ECM, whereas laminin (Figure 1r) was not strongly evident in the exocrine tissue but was found lining the ducts. 

### 3.2. Tissue-Specific Properties of Human Decellularized Scaffolds for PDAC Cultures

Following successful decellularization of pancreatic and liver tissue, the next step was to investigate its ability to engraft PDAC cells onto the scaffolds. Three cell lines were selected based on anatomical site of origin and metastatic potential; PANC-1 (primary non-metastatic cells isolated from the pancreas), MIA PaCa-2 (primary metastatic cells isolated from the pancreas), and PK-1 (metastatic cells isolated from the liver).

After 7 days in culture, PANC-1 cells were observed to attach to both pancreatic and liver scaffolds (Appendix A) without extended migration into the scaffolds. After 14 days, PANC-1 cells repopulating the pancreas scaffolds migrated deeper into the parenchymal space as well as the walls of the pancreatic ducts (Figure 2a). This was not observed in the liver scaffolds, where the cells attached superficially (Figure 2b). MIA PaCa-2 cells invaded both tissue types similarly. After 7 days, cells migrated as singular units into the parenchyma (Appendix A). After 14 days of culture, MIA PaCa-2 cells repopulating both pancreas (Figure 2c) and liver (Figure 2d) scaffolds presented a maintained presence as day 7. Finally, PK-1 cells showed similar cell behavior at both day 7 (Appendix A) and day 14 of culture (Figure 2e,f). However, there was a distinct difference in behavior between the liver and pancreas tissue. PK-1 cells engrafted the liver scaffold, migrated and attached to all major vessels as well as some smaller vessels (Figure 2f). This was not observed on the pancreatic scaffolds as the cells created thick layers on the outer surface of the scaffold, attaching as aggregates (Figure 2e).

To further investigate the significance of tissue-specificity, RNAseq was performed on PANC-1 and PK-1 cells cultured on both pancreatic and liver scaffolds. Differential expression (DE) analysis of PANC-1 cells cultured on liver scaffolds showed an upregulation of 773 genes and a down-regulation of 790 genes (Log2FC > 0.5, adjp < 0.05) when compared to those cultured on pancreatic scaffolds (Figure 2g). Functional enrichment analysis identified multiple gene sets associated with cell adhesion and the extracellular matrix (biological adhesion, cell–cell adhesion, extracellular matrix organization, extracellular structure organization) (Figure 2h). Furthermore, genes associated with the hallmark signature of epithelial to mesenchymal transition (EMT) were highly enriched (adjp < 0.0001). In particular, expression of CDH2 (encoding N-cadherin) was increased on the liver scaffolds, whereas expression of CDH1 (E-cadherin) was significantly decreased. This is consistent with a switch to a more mesenchymal phenotype induced by the liver ECM compared to the pancreas. 

DE analysis of PK-1 cells cultured on liver scaffolds showed an up-regulation of 684 genes and a down-regulation of 1590 genes (Log2FC > 0.5, adjp < 0.05) when compared to those cultured on pancreatic scaffolds (Figure 2i). As observed in PANC-1 cells, genes related to EMT were highly enriched (adjp < 0.0001), alongside gene sets associated with the extracellular matrix, tissue development and cell morphogenesis (adjp < 0.0001, adjp < 0.0001 and adjp < 0.0001, respectively). (Figure 2j). Gene sets related to hypoxia, extracellular structure organization, and biological adhesion were also enriched. Taken together, it appears that the liver scaffold induces genes (e.g., CDH2, LAMC2, ITGAV, ANPEP, CADM1, TGFBI, COL11A1, PTX3, VEGFC) known to drive a mesenchymal cell phenotype that is associated with metastatic-like behavior and the expression of genes involved in multiple aspects of remodeling of the ECM (e.g., FN1, ADAMTS15, MMP2, MMP9, TIMP2, TIMP3, SPOCK2). Inspection of the genes showing significantly up-regulated expression on the liver scaffold also identified various genes associated with pancreatic cancer. This included elements of the Hedgehog signaling pathway (SHH, GAS1), semaphorin signaling (SEMA3A, PLXNA1, PLXNB2, PLXND1, PLXNA2, FARP1, FARP2) and also axon guidance pathways (ROBO1, SLIT1, SLIT3 and SLITRK2), all notable for their involvement in pancreatic cancer progression and metastasis [29,30,31] (Figure 2k).

The association of genomically-defined subtypes of pancreatic cancer with patient survival and drug resistance has been proposed as a mechanism to stratify patients for personalized medicine approaches, with strong evidence being presented by the Precision-Panc project [32]. We queried whether the ECM-scaffold derived from pancreas or liver could alter the subtype of cancer cell lines cultured on them. Upregulated genes in cells cultured on liver scaffolds relative to pancreatic scaffolds were compared with the transcriptomes of PDAC samples classified according to the gene expression subtypes described by Moffitt et al. [33] from The Cancer Genome Atlas (TCGA) [34]. Using gene set enrichment analysis (GSEA), we observed a switch from the classical subtype [33] when cells were cultured on the pancreas ECM, to the basal subtype when cells were cultured on the liver ECM. This was observed for both PANC-1 and PK-1 cell lines (Figure 2l). 

To identify clinically relevant pathways correlated to aggressive pancreatic cancer cases, we determined which genes were significantly upregulated in tumours of patients who exhibited poor overall survival (lower quartile) versus those who performed better (upper quartile) (Figure 3a). It was observed that 1246 genes show significantly increased expression (Log2FC > 0.5, q-value < 0.25) in the lower quartile (median survival 5 months) versus the upper quartile (median survival 67 months) (Figure 3b). An overlap analysis of those genes significantly upregulated in PANC1 cells cultured on the liver scaffold compared to the pancreatic scaffolds identified 120 common genes (Figure 3c). STRING [27] network analysis and functional enrichment analysis identified enriched clusters of genes related to interferon alpha and gamma responses alongside EMT, extracellular matrix organization, collagen formation and integrin–cell surface interactions (Figure 3d).

### 3.3. Resistance of PDAC Cells to Chemotherapy Treatments in 3D Tissue-Specific ECM Scaffolds

To further investigate and validate our PDAC models for drug screening and chemoresistance studies, it was important to confirm the ability of the 3D PDAC models to mimic an in vivo-like behavior when treated with established drugs. To that effect, two chemotherapeutics were chosen: (i) gemcitabine, the most acknowledged chemotherapy for PDAC patients and (ii) doxorubicin, one of the most potent and widely used chemotherapeutic agents, although not currently applied in PDAC management.

The three cell lines, PANC-1, MIA PaCa-2, and PK-1, were first cultured in 2D and a series of nine different concentrations for gemcitabine (0.01 µM to 0.5 µM) and for doxorubicin (0.001 µM to 0.8 µM) were tested (Appendix A). Cell survival was measured by Alamar Blue assay 96 h post-treatment and quantified as a percentage of change in respect to untreated control cells (media with no treatment). PANC-1 cells treated with gemcitabine presented no substantial cell death with concentrations lower than 0.1 µM. Cell survival was at ~60% at the maximum dose used of 0.5 µM (Appendix A, blue line). A similar trend was observed for doxorubicin treatment, which showed no substantial change in cell death from 0.1 µM to 0.5 µM although a significant increase was present between 0.5 and 0.8 µM (Appendix A, blue line). The viability of MIA PaCa-2 cells was not significantly affected beyond 0.2 µM gemcitabine, but cell survival was at ~15% at the maximum dose used of 0.5 µM (Appendix A, green line). This was also very similar for cells treated with doxorubicin, which showed no substantial cell death beyond 0.05 µM between the incremental drug concentrations, reaching a cell survival of ~10% at a concentration of 0.8 µM (Appendix A, green line). PK-1 cells presented no substantial cell death with gemcitabine beyond 0.2 µM between the incremental drug concentrations, and cell survival was at ~20% at the maximum dose of 0.5 µM (Appendix A, red line). Although, when treated with doxorubicin, cell death kept increasing significantly in a dose-dependent manner, reaching a cell survival of ~15% at the concentration of 0.8 µM (Appendix A, red line).

Reflecting on their effect in 2D, it was decided, for consistency, to use 0.5 µM for both gemcitabine and doxorubicin on all the 3D model experiments (PANC-1, PK-1 and MIA PaCa-2 on both pancreas and liver scaffolds). Similar to the 2D culture experiments, cell survival was measured using Alamar Blue and quantified as a percentage of the untreated control group (media with no treatment).

PANC-1 cells treated with gemcitabine showed no significant change in cell survival between the pancreas and liver scaffolds (*p* > 0.05), whereas a significant reduction was observed in 2D culture cell survival of 31.10 ± 2.06% in comparison to pancreas scaffolds (*p* < 0.0001) and a significant reduction of 31.46 ± 2.98% in comparison to liver scaffolds (*p* < 0.0001) (Figure 4a). PANC-1 cells treated with doxorubicin had a significant reduction of 10.98 ± 3.03% in cell survival in the pancreas scaffolds compared to liver scaffolds (*p* < 0.001). Additionally, there was a significant reduction in 2D culture cell survival of 38.54 ± 1.36% in comparison to pancreas scaffolds (*p* < 0.0001) and a significant reduction of 49.52 ± 2.75% in comparison to liver scaffolds (*p* < 0.0001) (Figure 4a). 

MIA PaCa-2 cells treated with gemcitabine had no significant change in cell survival between the pancreas and liver scaffolds (*p* > 0.05) whereas a significant reduction was observed in cell survival of 37.09 ± 1.56% on 2D plastic in comparison to pancreas scaffolds (*p* < 0.0001) and a significant reduction of 45.64 ± 2.40% was observed in comparison to liver scaffolds (*p* < 0.0001) (Figure 4b). MIA PaCa-2 cells treated with doxorubicin had no significant change in cell survival between the pancreas and liver scaffolds (*p* > 0.05) whereas a significant reduction was observed in cell survival of 50.93 ± 2.21% in 2D culture in comparison to pancreas scaffolds (*p* < 0.0001) and a significant reduction of 54.07 ± 1.95% was observed in comparison to liver scaffolds (*p* < 0.0001) (Figure 4b).

PK-1 cells treated with gemcitabine had a significant reduction of 29.58 ± 5.00% in cell survival in the liver scaffolds compared to pancreas scaffolds (*p* < 0.001). Additionally, there was a significant reduction in cell survival of 64.74 ± 2.99% in 2D culture in comparison to pancreas scaffolds (*p* < 0.0001) and a significant reduction of 35.16% ± 4.07% in comparison to liver scaffolds (*p* < 0.0001) (Figure 4c). PK-1 cells treated with doxorubicin had a significant reduction of 21.93 ± 1.86% in cell survival in the pancreas scaffolds compared to liver scaffolds (*p* < 0.001). Additionally, a significant reduction in cell survival of 52.99 ± 1.79% on 2D plastic in comparison to pancreas scaffolds (*p* < 0.0001) and a significant reduction of 74.92 ± 0.77% in comparison to liver scaffolds (*p* < 0.0001) (Figure 4c) were observed.

To prove that chemotherapy drugs were able to reach and be taken up by the cells growing into the scaffolds, detection of the natural fluorescence of doxorubicin was utilized. PANC-1, MIA PaCa-2 and PK-1 cells cultured on pancreas scaffolds were treated with 1 µM doxorubicin for 24 h. The samples were then directly frozen and cryo-sectioned. As a control, untreated samples were also imaged. Control PANC-1 (Figure 4d top panel), MIA PaCa-2 (Figure 4e; top panel) and PK-1 (Figure 4f; top panel) samples did not present any fluorescence in the doxorubicin channel (TRITC; red) but did present a positive signal for nuclei (DAPI; blue) and collagen (FITC; green). On the other hand, in all the treated samples, PANC-1 (Figure 4d; bottom panel), MIA PaCa-2 (Figure 4e; bottom panel) and PK-1 (Figure 4f; bottom panel) doxorubicin fluorescence (TRITC; red) was detected, and nuclei (DAPI; blue) and collagen (FITC, green) were observed as well. When the three channels were merged, it was evident that doxorubicin had reached the nuclei in the 3 cell types employed (MERGE; purple).

### 3.4. Chemoresistance Characteristics of PDAC Cells

To further investigate the effect of chemotherapy on PDAC cells further, immunohistochemical staining for γH2A (marker of DNA damage) and ASP175/cleaved caspase 3 (marker of apoptosis) was performed. PANC-1 cells on pancreas scaffolds presented negative staining for γH2A in non-treated control samples but were positive on the gemcitabine-treated samples. Additionally, a high intensity of staining was present on the doxorubicin-treated samples (Figure 5a; top panel). ASP175 staining was also negative in the control samples, which was similarly observed in the majority of cells in the doxorubicin (Figure 5a; bottom panel)-treated samples, whereas a higher number of cells stained positive in the gemcitabine group (Figure 5a; bottom panel)-treated samples. PANC-1 cells on liver scaffolds presented negative staining for γH2A (Figure 5b; top panel) and ASP175 (Figure 5b; bottom panel) in all conditions: control, gemcitabine and doxorubicin. Therefore, these results suggest that both chemotherapies caused DNA damage in PANC-1 cells cultured on pancreas scaffolds, but the cells managed to avoid apoptosis. On the contrary, the same chemotherapies did not result in DNA damage to the PANC-1 cells repopulating the liver scaffolds, indicating a higher chemoresistance.

MIA PaCa-2 cells on pancreas scaffolds did not express γH2A in the control samples but were positive in gemcitabine-and doxorubicin-treated samples (Figure 5c; top panel). ASP175 staining was negative in the control samples, and in the majority of cells when treated with gemcitabine and doxorubicin (Figure 5c; bottom panel). MIA PaCa-2 cells on liver scaffolds did not express γH2A (Figure 5d; top panel) and ASP175 (Figure 5d; bottom panel) in any of the tested conditions: control, gemcitabine or doxorubicin. Therefore, similar to the PANC-1 results, both chemotherapies caused DNA damage in the MIA PaCa-2 cells in pancreas scaffolds, but the cells managed to avoid apoptosis. Similar to the PANC-1 cells, the same chemotherapy approaches did not result in DNA damage and cell death of MIA PaCa-2 cells repopulating the liver scaffolds.

PK-1 cells on pancreas scaffolds stained for γH2A were negative in the control samples but were faintly positive upon gemcitabine treatment and a moderate intensity of staining was observed in the doxorubicin-treated samples (Figure 5e; top panel). ASP175 staining was negative in all conditions (Figure 5e; bottom panel). PK-1 cells on liver scaffolds did not express γH2A (Figure 5f; top panel) and ASP175 (Figure 5f; bottom panel) in any tested conditions. Therefore, for PK-1 cells in pancreas scaffolds, both chemotherapies caused less DNA damage in comparison to the other cell lines, and similarly, the cells managed to avoid apoptosis. The same chemotherapy treatments did not result in DNA damage to the PK-1 cells populating the liver scaffolds.

To examine the influence of the ECM on chemoresistance, expression of mRNA from gemcitabine-treated PANC-1 cells cultured on pancreas scaffolds and PK-1 cells cultured on liver scaffolds was analyzed using RNA sequencing. In PANC-1 cells, 2219 genes were significantly induced (Log2FC > 0.5, p-Adj < 0.05) and 981 genes showed decreased expression after therapy (Figure 6a). A similar effect was observed in PK-1 cells, where 1357 genes were significantly up-regulated by gemcitabine treatment and 651 showed decreased expression (Figure 6b). Functional enrichment analysis identified many enriched datasets in both cell lines associated with DNA replication, cell cycle, mitosis and the G2/M cell cycle checkpoint (E2F targets, DNA replication, cell cycle, G2M checkpoint, mitotic cell cycle), which is consistent with the known effects of gemcitabine and DNA damage on cell cycle progression (Figure 6c,d). This also demonstrated that gemcitabine was able to reach the cells within the scaffold and elicit the expected effects on the cell cycle and DNA replication. To query why the cells exhibited resistance to gemcitabine on the scaffolds, the expression of a panel of genes known to be associated with resistance to gemcitabine was assessed (CDT1, FOXM1, RRM1, RRM2, DNMT1, SNAI1, ISG15 and CDA) (12, 26–32). All the genes showed significantly increased expression following gemcitabine treatment (Log2FC > 0.5, p-Adj < 0.05) with the exception of CDA in the PK-1 cell line (Figure 6e,f). This suggests that multiple resistance mechanisms are triggered by cells, thus significantly blunting the effectiveness of gemcitabine on cells cultured in the ECM scaffolds. To explore the clinical significance of these genes, we looked again for overlap with genes significantly overexpressed in PDAC patients with low overall survival. Out of 2219 genes induced by gemcitabine in PANC1 cells on the pancreas scaffold, 330 genes were also overexpressed in the tumours of patients with low overall survival (Figure 7a). For PK1 cells on the liver scaffolds, out of 1357 genes induced by gemcitabine, 242 were also associated with low overall survival (Figure 7b). Functional analysis demonstrated that gene sets related to cell cycle regulation, mitosis and DNA replication were enriched in both cell lines (Figure 7c,d). Notably, the transcription factor FOXM1 was significantly upregulated. FOXM1 is a highly potent driver of cell cycle progression and has been associated with both a poor prognosis in PDAC patients and resistance to gemcitabine [35]. This suggests that a feature of both gemcitabine-treated cells and aggressive tumours is elevated expression of cell cycle control genes. 

## 4. Discussion

Despite advances in our understanding of the molecular pathogenesis and pathological progression of PDAC, no significant improvements in patient survival rates have been achieved since the early 1970s [29]. Several characteristics of PDAC, including its late onset of symptoms, propensity to metastasize at an early disease stage, resistance to chemo- and radiotherapy, along with a unique stromal niche, make this disease the fourth leading cause of cancer death in Western countries [30]. PDAC is largely characterized by deposition of ECM and stromal compartment activation as part of the desmoplastic reaction. This process drives biochemical and biomechanical effects on PDAC cells, and is one of the key mechanisms leading to chemo- and radiotherapy resistance [18].

To create better in vitro models for studying PDAC that resemble its behavior and characteristics in humans, this study has focused on the application of tissue-specific human ECM scaffolds to gain a better understanding of the role of the tissue microenvironment in modulating metastasis and chemoresistance in PDAC. 

Although the importance of the ECM in modulating several processes, including the physical properties of solid tumours, are well characterized (37), its bioactive role, which is mediated through biochemical and biomechanical properties, in modulating cellular phenotype and the direct contribution to tumour progression and chemotherapy resistance has often been overlooked. In recent years, an emerging technique to decellularized tissues has been developed and optimized to obtain and study the ECM, thus providing a detailed picture of the heterogeneity of tissue-specific acellular microenvironments [19,20,31,32]. The development of ECM scaffolds as a platform for 3D in vitro cell culture is of primary importance, as it can provide high-quality acellular scaffolds with preserved ECM components and structure to recreate the native environment to promote cell engraftment. Our results suggest that the highly preserved and unique composition of the ECM plays a bioactive role in regulating PDAC progression and chemotherapy response. Indeed, PANC-1 (primary PDAC cell line) and PK-1 (metastatic PDAC cell line) cells showed different migratory behaviors on pancreas and liver scaffolds, whereas MIA PaCa-2 cells (primary PDAC cell line with metastatic capability) had similar migration patterns on both scaffolds. These results, which are in agreement with findings from other groups [33,34,35], suggest that the ECM plays a key role in modulating cell phenotype, both ex vivo and in vivo. Next-generation sequencing of PANC-1 and PK-1 cells repopulated on pancreas and liver scaffolds, performed in both tissue-matched and tissue-mismatched experiments, identified key biological steps involved in the progression from PDAC to liver metastasis. These findings provide a novel insight of a “tumour-ECM niche” dictated by tissue-specific acellular environments. Our data also demonstrate the ability of the liver ECM scaffold to promote the expression of the key genes/pathways associated with pancreatic cancer development. For example, expression of *SHH* and *GAS1*, encoding ligands in the Hedgehog pathway, has been implicated in the activation of stromal cells and the development of the desmoplastic environment [36,37]. Multiple genes in the semaphorin pathway (*SEMA3A, SEMA3C*), plexin and neuropilin receptors (*PLXNA1*, *PLXNA2*, *PLXNB2*, *PLXND1*) and downstream genes (*FARP1*, *FARP2*), known to activate *RAC1* and *FLNA,* promoting invasion and metastasis, were also upregulated, some of which have been associated with pancreatic cancer and poor survival [38]. Furthermore, our findings on how the ECM scaffold can mediate subtype-switching from the classical type on the pancreas to the basal type on the liver, raise a key consideration for therapeutic strategies, which may be driven by cancer tissue biopsy profiling from a single site. If the metastatic site promotes a different subtype to the primary site, the selection of a therapy may only be appropriate for the primary tumour and have reduced or no efficacy towards the metastatic disease. In this case, combination or sequential/alternating therapies may be more effective. We also looked at the overlap of genes overexpressed in the scaffold model of early metastasis (PANC-1 cells on the liver ECM versus the pancreas ECM) and genes overexpressed in PDAC tumour samples from patients with poor survival rates. This highlighted gene sets related to the response to interferon alpha and gamma, EMT and various gene sets related to interaction with/formation of the ECM. Notably, a signature of interferon response genes has been associated with poor survival of PDAC patients. It has also been shown to result in reprogramming the stromal environment to promote tumour growth [39]. This finding was attributed to low levels of DNA methylation, resulting in higher expression of endogenous retroviral transcripts. Consequently, inhibition of interferon gene expression by the STAT1 inhibitor ruxolitinib has been demonstrated as a potential therapeutic approach in preclinical models of pancreatic cancer, with suggested utility in patients with a high interferon gene signature [40]. Therefore, our data show how the extracellular environment may influence cellular transcriptional programs, particularly in the setting of metastatic disease and how a decellularized scaffold model may identify clinically relevant mechanisms that drive aggressive disease.

The mechanisms orchestrating chemotherapy resistance in PDAC and other solid tumours are mainly driven by genomic instability and oncogene activation, as well as by the physical barrier provided by the ECM in limiting tumour perfusion and the delivery of anti-tumour drugs [18]. In line with the work described herein, we further assessed whether the ECM could drive chemotherapy resistance, not only through physical impairment in drug permeability but also through its bioactive role in promoting cellular resistance mechanisms. 

To this end, primary (PANC-1 on pancreas scaffolds) and metastatic (PK-1 on liver scaffolds) PDAC models were used to examine tissue-specific chemoresistance to gemcitabine. In line with previous results, viability assays showed an increase of chemoresistance in the 3D models in comparison to the 2D cultures [41,42,43,44,45]. In order to assess whether the reported chemoresistance was an artificial effect due to the lack of drug penetration through the ECM models, the engineered cancer models were dissected, and the presence of fluorescent chemotherapy (doxorubicin) was assessed. Notably, PDAC and liver metastatic cell lines were resistant to chemotherapy at the examined time points, even though the drug was found to reach the cell’s nucleus within the first 24 h. This finding suggests the feasibility for therapeutics to diffuse through highly dense ECM fibers as well as the possibility of non-multi drug resistance mechanisms to occur. To investigate this further, RNAseq was used to compare gemcitabine-treated and untreated (control) samples on 3D scaffold models. A number of genes associated with gemcitabine resistance were found to be upregulated following treatment of cells on the ECM scaffolds (*CDT1*, *FOXM1*, *RRM1*, *RRM2*, *DNMT1*, *SNAI1*, *ISG15* and *CDA*). In addition, the adoption of an EMT-signature on the liver scaffold has also been shown to confer resistance to therapy [31]. Therefore, both CDA-mediated gemcitabine degradation and excretion [46] and ribonucleotide reductase-mediated DNA synthesis [47] may contribute to the resistance observed. Furthermore, many genes involved in cell cycle progression were also upregulated in response to treatment that may have allowed PDAC cells to bypass apoptosis and progress through the cell cycle despite the DNA damage caused by gemcitabine. These genes were also overexpressed in the tumours of PDAC patients showing poor overall survival. Thus, it raises the possibility that chemotherapy may induce a selection of chemotherapy-resistant cells within their natural acellular environment, for which an alternative therapeutic strategy may be required. Additionally, the expression of several cell adhesion molecules, as well as cell motility and ECM regulators were found to be altered in the treated samples. 

## 5. Conclusions

In conclusion, these results suggest that tissue-specific ECM provides the tumour-ECM niche “cues” for primary and metastatic PDAC cells. Moreover, there is an evident alteration in cell response to cancer therapies in the presence of a tissue-specific ECM niche. These observations reinforce the need to develop more complex in vitro models including the acellular microenvironment in solid tumours for (i) identifying novel therapeutic targets, (ii) screening therapeutics, and (iii) discovering biomarkers. Future studies will require the inclusion of other components to better recapitulate the complexity of solid tumours, including stromal and immune cells, as well as micro-perfusion systems.

## 6. Patents

The following patent resulted from this work:

■Human pancreas scaffolds.
○Patent Number: EP3613447B1.○Status: Granted


## Figures and Tables

**Figure 1 cells-11-03652-f001:**
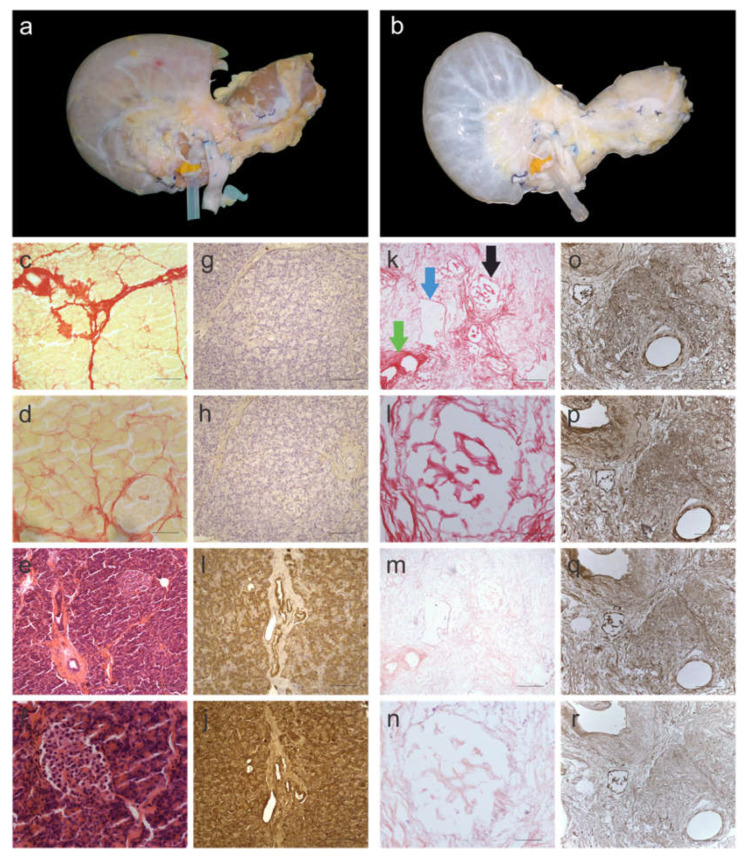
Characterization of decellularized human pancreas tissue. (**a**) before and (**b**) after decellularization. Yellow/pink tissue is a representative of cellular material, which is the colour of fresh pancreata, whereas the translucent-white color is a representative of decellularized tissue. Histological images of native pancreas stained with Picro-Sirius Red (SR) and imaged at (**c**) 10× and (**d**) 40× magnification. SR stains cellular material yellow and collagen fibres red. Histological images of native pancreas stained with H&E and imaged at (**e**) 10× and (**f**) 40× magnification. H&E stains nuclear material blue and the ECM red. Immunohistology images of native pancreas stained for (**g**) collagen I, (**h**) collagen III, (**i**) fibronectin and (**j**) laminin. Histological images of decellularized pancreas stained with SR and imaged at (**k**) 10× and (**l**) 40× magnification. SR staining showed no evidence of cellular material. Histological images of decellularized pancreas stained with H&E and imaged at (**m**) 10× and (**n**) 40× magnification. H&E staining showed no evidence of nuclear material. Immunohistology images of decellularized pancreas stained for (**o**) collagen I, (**p**) collagen III, (**q**) fibronectin and (**r**) laminin. The black arrow is pointing at an islet of Langerhans, the green arrow is pointing at an exocrine ECM duct and the blue arrow is pointing at a blood vessel, which were well maintained in decellularized pancreata. Scale bars for (**c**,**k**,**e**,**m**) are 200 µm, for (**d**,**f**,**l**,**n**) are 50 µm and for (**g**–**j**,**o**–**r**) are 100 µm. (*n* = 6 pancreata).

**Figure 2 cells-11-03652-f002:**
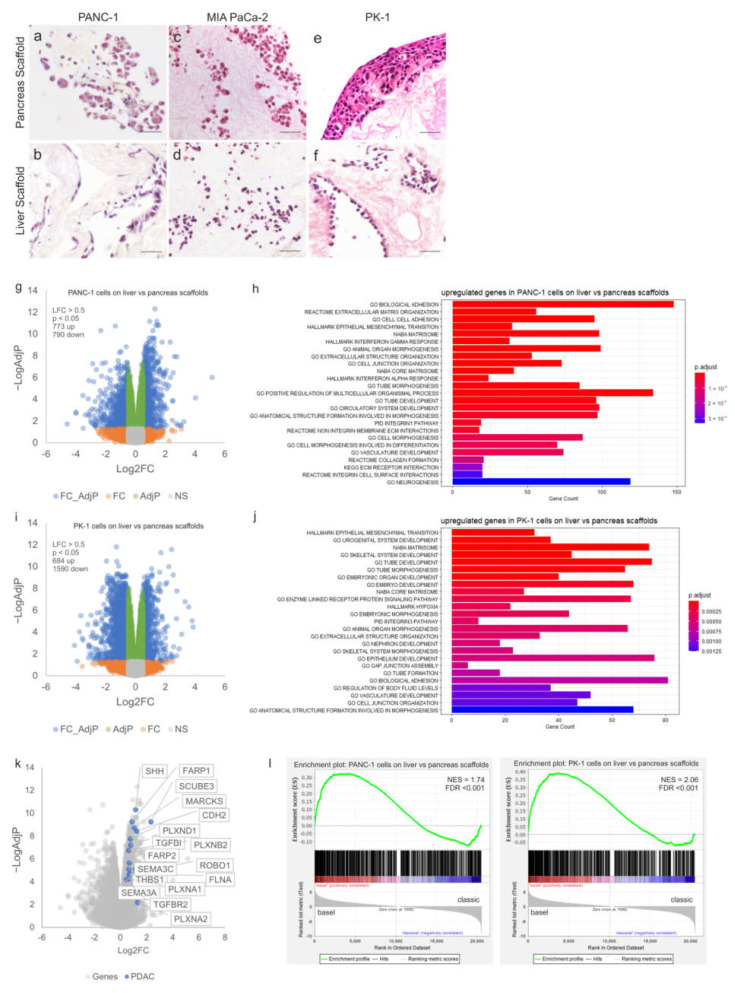
Tissue-specific properties of human decellularized scaffolds for PDAC cultures. H&E staining of PDAC cells cultured for 14 days on decellularized scaffolds showed (**a**) PANC-1 cells on pancreas scaffolds migrated into the ECM and (**b**) PANC-1 cells on liver scaffolds only attached on the edge of the ECM with no migration inward. (**c**) MIA PaCa-2 cells on pancreas scaffolds migrated into the ECM and (**d**) MIA PaCa-2 cells on liver scaffolds also migrated into the ECM. (**e**) PK-1 cells on pancreas scaffolds grew into thick clusters on the edge of the ECM and (**f**) PK-1 cells on liver scaffolds invaded the vessels and migrated into the ECM as thick clusters. Volcano plot visualizing the gross gene expression changes as determined by RNAseq comparing (**g**) PANC-1 cells on liver scaffolds versus pancreas scaffolds and (**i**) PK-1 cells on liver scaffolds versus pancreas scaffolds. Genes that showed significantly altered expression (LFC > 0.5, *p* < 0.05) are indicated in blue. Functional enrichment analysis of the differentially overexpressed genes with the lowest adjusted *p*-values between cells cultured on liver and pancreatic scaffolds is shown, (**h**,**j**) depict the results for PANC-1 and PK-1 cells, respectively. Bar length is analogous to the number of differentially expressed genes in the functional annotation category, while the bar color depicts the adjusted *p*-value for the hypergeometric test performed. (**k**) Volcano plot visualizing the gene expression changes as determined by RNAseq comparing PANC-1 cells on liver scaffolds versus pancreas scaffolds with significantly overexpressed genes (LFC > 0.5, *p* < 0.05) known to play a role in pancreatic cancer and metastasis highlighted. (**l**) GSEA plots showing enrichment of upregulated genes on liver versus pancreas scaffolds in basal compared to classical Moffit PDAC subtypes from the TCGA. NES, normalized enrichment score; FDR, false discovery rate. Scale bars for (**a**–**f**) are 50 µm. (*n* = 8 per condition for (**a**–**f**)), and (*n* = 3 per condition (**g**–**l**)).

**Figure 3 cells-11-03652-f003:**
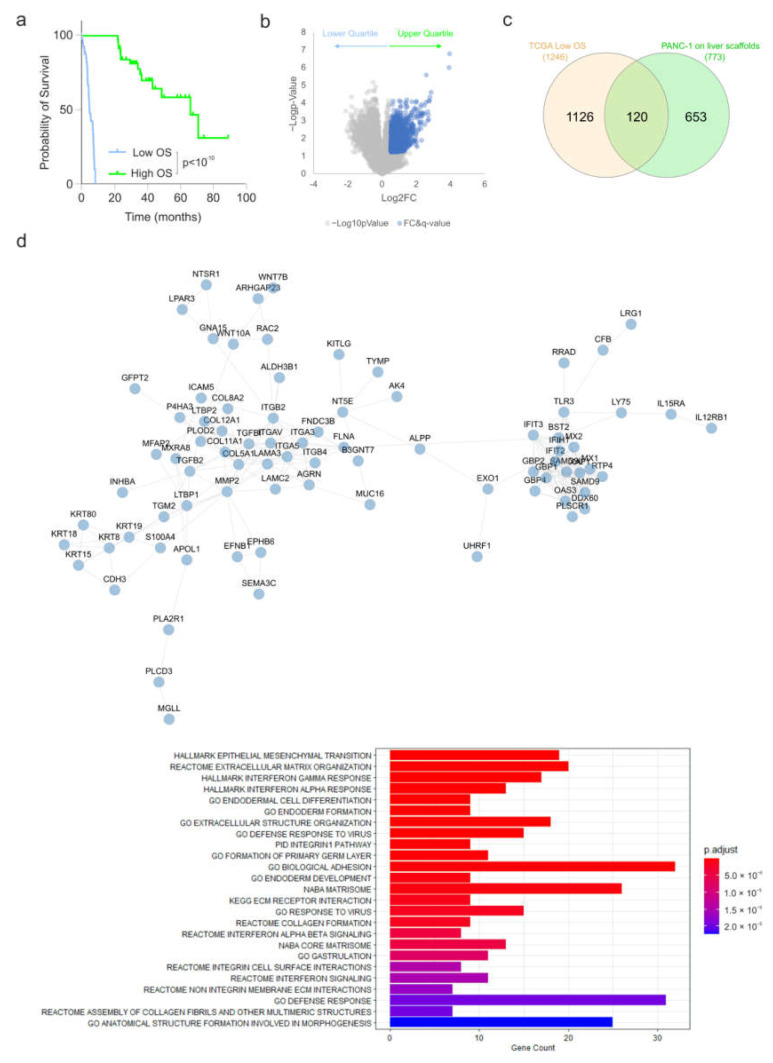
Identification of genes induced in PANC1 cells on liver scaffolds and those associated with clinically aggressive PDAC. (**a**) Overall survival (OS) of the upper and lower quartiles of 184 PDAC patients from the TCGA PanCancer Atlas (cBioPortal). (**b**) Volcano plot of gene expression for patient tumours in the upper and lower quartiles for overall survival from (**a**). Genes highlighted in blue indicate those showing significantly increased expression (LFC > 0.5, q-value < 0.25). (**c**) Overlap analysis of genes increased in PANC-1 cells on the liver scaffold compared to the pancreas scaffold and those increased in the tumours of patients displaying poor survival. (**d**) STRING interaction network for the 120 genes identified in the overlap analysis. Functional enrichment analysis of the 120 genes identified in the overlap analysis is presented. Bar length is analogous to the number of differentially expressed genes in the functional annotation category, while the bar color depicts the adjusted *p*-value for the hypergeometric test performed.

**Figure 4 cells-11-03652-f004:**
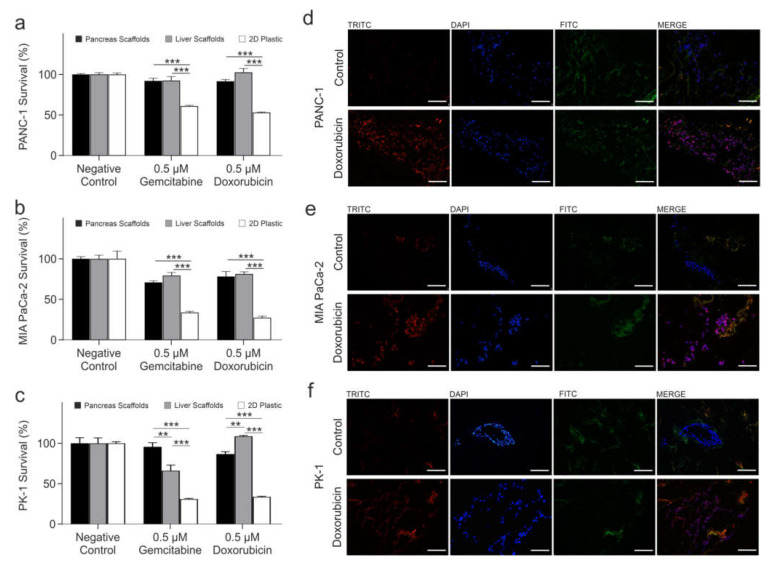
Comparison of PDAC cell response to chemotherapeutic treatments in 3D scaffolds vs. 2D plastic. Alamar blue viability assay of (**a**) PANC-1, (**b**) MIA PaCa-2 and (**c**) PK-1 cells cultured on pancreas scaffolds (black bars), liver scaffolds (grey bars) and 2D plastic, treated with 0.5 μM gemcitabine and 0.5 μM doxorubicin. Validation of doxorubicin uptake by PDAC cells. Fluorescent imaging of (**d**) PANC-1, (**e**) MIA PaCa-2 and (**f**) PK-1 cultured on pancreas scaffolds that were not treated (control; top panel) did not present any fluorescence in the doxorubicin channel (TRITC; red) but presented a positive fluorescence for nuclei (DAPI; blue) and collagens (FITC; green). All doxorubicin-treated samples (bottom panel) presented a fluorescence on the doxorubicin channel (TRIRC; red), nucleus channel (DAPI, blue) and collagen channel (FITC, green). When the three channels were merged, it was evident that doxorubicin had reached the nucleus (purple). All the images were obtained using a 20x objective. Scale bars are representative of 100 μm. Data are expressed as mean ± s.d. ** *p* < 0.001 and *** *p* < 0.0001 (*n* = 8 scaffolds per condition for (**a**–**c**)) and (*n* = 3 scaffolds for (**d**–**f**)).

**Figure 5 cells-11-03652-f005:**
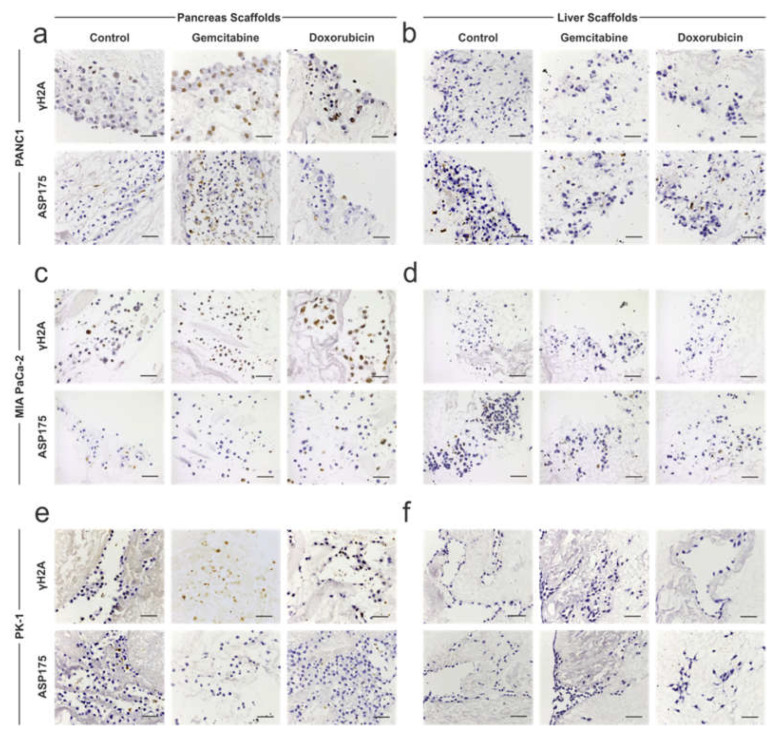
Immunohistochemistry analysis of chemotherapy-treated PDAC cells cultured on 3D scaffolds. PDAC cells cultured on 3D scaffolds were left untreated (control) or treated with 0.5 μM gemcitabine and 0.5 μM doxorubicin and stained for γH2A (top panels) or ASP175 (bottom panels). (**a**) γH2A staining of PANC-1 cells on pancreas scaffolds stained negative on the control samples but were positive on the gemcitabine- and doxorubicin-treated samples. There was no difference in staining of ASP175 between the different conditions. (**b**) γH2A and ASP175 staining of PANC-1 cells on liver scaffolds were negative in all conditions. (**c**) MIA PaCa-2 cells on pancreas scaffolds stained for γH2A were negative in the control samples but was positive in the gemcitabine- and doxorubicin-treated samples. There was no difference in staining of ASP175 between the different conditions. (**d**) MIA PaCa-2 cells on liver scaffolds stained for γH2A and ASP175 were negative in all conditions. (**e**) PK-1 cells on pancreas scaffolds stained for γH2A were negative in the control sample but was faintly positive in the gemcitabine-treated sample and moderately positive in the doxorubicin-treated sample. All conditions showed a negative staining for ASP175. (**f**) PK-1 cells on liver scaffolds, stained for γH2A and ASP175, were negative in all conditions. Scale bars are representative of 100 μm.

**Figure 6 cells-11-03652-f006:**
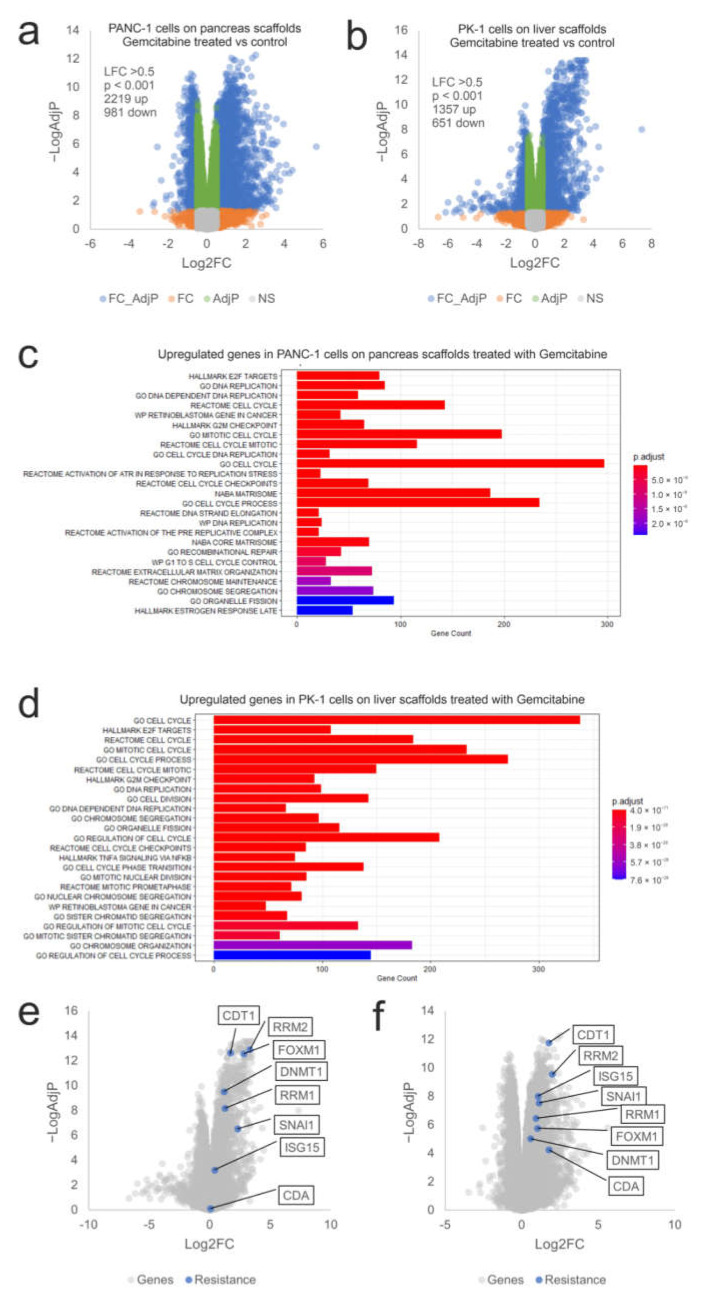
Transcriptional response of PANC-1 and PK-1 cells treated with gemcitabine on their native scaffolds. Volcano plots of (**a**) PANC-1 and (**b**) PK-1 cells treated with 0.5 µM gemcitabine on their native scaffolds. Blue symbols represent genes showing significantly altered expression (LFC > 0.5, *p* < 0.05). Functional analysis of the overexpressed genes identified in (**c**) PANC-1 and (**d**) PK-1 cells treated with gemcitabine. Expression of a panel of genes related to gemcitabine resistance in (**e**) PANC-1 and (**f**) PK-1 cells are indicated in blue. (*n* = 3 per condition).

**Figure 7 cells-11-03652-f007:**
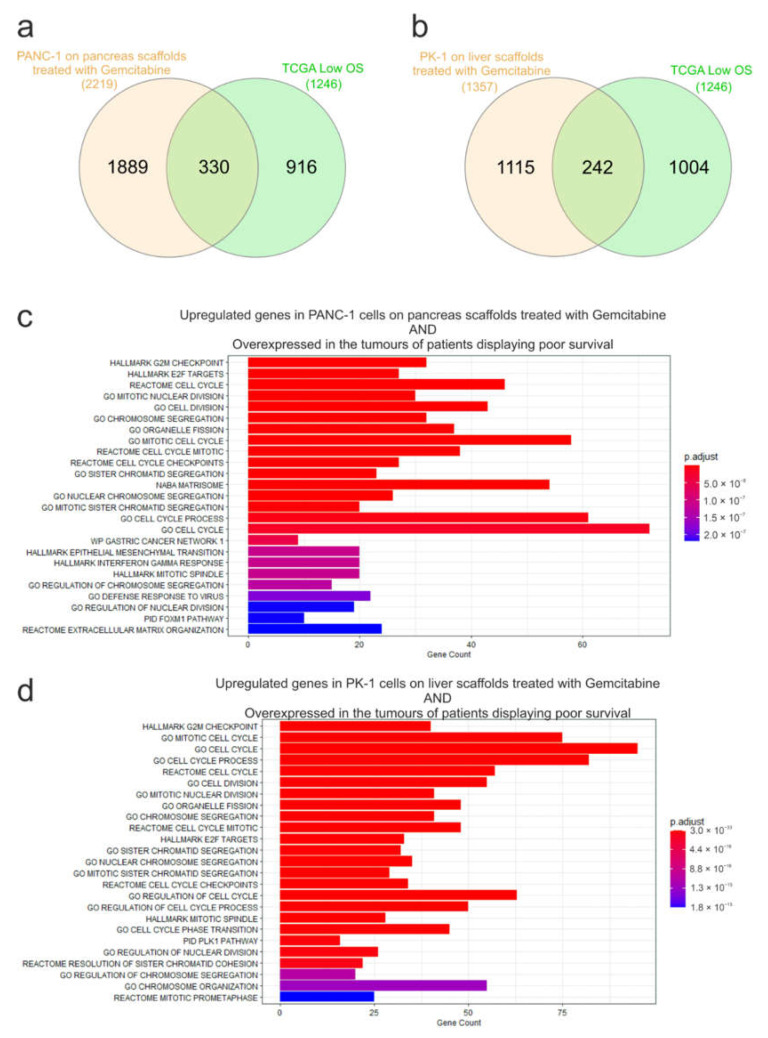
Overlap analysis of genes significantly induced by gemcitabine treatment and genes associated with low overall survival in pancreatic cancer patients. Venn diagrams to show the overlap of genes significantly overexpressed in (**a**) PANC-1 or (**b**) PK-1 cells following gemcitabine treatment and those increased in the tumours of patients displaying poor survival i.e., low overall survival (OS). Functional analysis of the genes identified in (**c**) PANC-1 and (**d**) PK-1 cells treated with gemcitabine and also found to be overexpressed in the tumours of patients displaying poor survival.

## Data Availability

Not applicable.

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
