# Peer review of "Tissue-Specific Human Extracellular Matrix Scaffolds Promote Pancreatic Tumour Progression and Chemotherapy Resistance"

_cells, 2022, doi:10.3390/cells11223652_

Round 1
Reviewer 1 Report
The research is devoted to the study of pancreatic ductal adenocarcinoma (PDAC), a highly aggressive cancer which is characterized by limited response to treatments.
The need in complex in vitro 3D models to study the disease exists. Authors offered in vitro 3D models of PDAC based on decellularized human pancreata either liver seeded with both primary (PANC-1 and MIA PaCa-2) and metastatic (PK-1) pancreatic cancer cells as more reliable in vitro cell culture. It is interesting that not only pancreatic decellularized fragments were picked as matrix for studying PDAC but also decellularized liver cubes, as a common PDAC metastatic sites, with analogous ECM components, had been chosen as tissue-specific extracellular matrix scaffolds. This approach allowed to compare and specify the effect of different type extracellular matrix scaffolds on cellular behavior including cell response to different cancer treatments. Authors demonstrated influence of 3D models microenvironment on regulating PDAC progression and chemotherapy response in comparison with 2D models and provided evidence that tissue-specific acellular environment impacted primary and metastatic PDAC cells.
The experiment is well-planned, and great amount of work had been done, the stages of the study are consistently characterized and described in details.
The scheme of the research which is presented in the Article clearly illustrates its stages and the idea of the experiment.
The methods and approaches are described properly.
It would be quite interesting to know the peculiarities of even more complicated in vitro 3D models with stromal or immune cells as components which Authors proposed for the future research.
The following comments do not diminish the value of the Article:
Line 333 Probably it would be better to rename the title of the item “This Decellularisation of human pancreata”?
Line 373 It would be better to place Figure 1 and its caption on the same page
Line 411 If it is possible, it would be better to increase the font size of the text in the Figure 2 to make the valuable information visible
Line 429-433 It would be better to place Figure 2 and its caption on the same page
Line 491 If it is possible, it would be better to increase the font size of the text in the Figure 3 to make the valuable information visible
Line 505 Probably it would be better to include the information about resistance of PDAC cells to chemotherapy treatments in 3D decellularised scaffolds into a separate item (subsection)? But it is very much upon Authors consideration
Line 690 If it is possible, it would be better to increase the font size of the text in the Figure 6 to make the valuable information visible
Line 701 It would be better to place Figure 7 and its caption on the same page
Author Response
We thank the reviewer for the positive feedback and the suggestions for the clarification and improvement of the manuscript. Below are the detailed, point-by-point responses to the reviewer’s comments.
Response to Comments and Suggestions for Authors:
Line 333 Probably it would be better to rename the title of the item “This Decellularisation of human pancreata”?
We thank the reviewer for the suggestion. We have now changed the title to “Characterisation of the decellularised human pancreata”
Line 505 Probably it would be better to include the information about resistance of PDAC cells to chemotherapy treatments in 3D decellularised scaffolds into a separate item (subsection)? But it is very much upon Authors consideration
We thank and agree with the reviewer. We have now added a subsection title as follows “3.3. Resistance of PDAC cells to chemotherapy treatments in 3D tissue-specific ECM scaffolds”
Line 373 It would be better to place Figure 1 and its caption on the same page
Line 411 If it is possible, it would be better to increase the font size of the text in the Figure 2 to make the valuable information visible
Line 429-433 It would be better to place Figure 2 and its caption on the same page
Line 491 If it is possible, it would be better to increase the font size of the text in the Figure 3 to make the valuable information visible
Line 690 If it is possible, it would be better to increase the font size of the text in the Figure 6 to make the valuable information visible
Line 701 It would be better to place Figure 7 and its caption on the same page
We thank the reviewer for the positive suggestion addressed in the above comments. We would like to highlight that the formatting of the text and figures follows a template supplied by Cells. We will contact the editorial office to ensure this will be changed in the published version of the manuscript.
Reviewer 2 Report
The manuscript aims to prove an unexpected assumption made by its authors, namely, the possible use of positive effects of tissue-specific scaffolds from decellularized human pancreas and liver on tumor progression and metastasis for the creation of in vitro PDAC models.
Comments and questions.
1. In the introduction, it should be clarified as to which data this assumption is based on, since there is a number of works showing the prospects of using tissue-specific scaffolds in the regenerative medicine and tissue engineering technologies, including stimulating the processes of the internal regeneration of damaged liver and pancreatic tissues.
2 2. Can the culturing of PDAC cells in 3D culture contribute to their better survival after a chemotherapeutic treatment versus a 2D culturing on plastic?
3 3. In the “section 2.12. Treatment of 3D Bioengineered Scaffolds with Chemotherapeutics” experiment, the PDAC cells on plastic in 2D culture were used as a control. In my opinion, it is not entirely correct to compare the survival parameters in 2D and 3D culturing conditions. It is better to use the same culturing conditions for comparison, in this case, in 3D culture.
4 4. Has the data on the survival of the PDAC cells following the chemotherapeutic treatment been compared with the results of the chemoresistance of these cells on scaffolds? For example, after the hemocitabine treatment of PK1 cells cultured on a scaffold from liver, DNA damage (yH2A-) and apoptosis processes (ASP175-) were not detected. However, significant cell death was noted (lines 561-564) in this treatment variant.
5 5. Based on the assumption made by the authors, there is a risk of developing oncological processes during the implantation of tissue-specific liver and pancreas scaffolds as part of tissue-engineered structures into damaged organs. Or does this assumption apply only to tissue tumors?
Author Response
We thank the reviewer for the positive feedback and the suggestions for the clarification and improvement of the manuscript. Below are the detailed, point-by-point responses to the reviewer’s comments.
Response to Comments and Suggestions for Authors:
- In the introduction, it should be clarified as to which data this assumption is based on, since there is a number of works showing the prospects of using tissue-specific scaffolds in the regenerative medicine and tissue engineering technologies, including stimulating the processes of the internal regeneration of damaged liver and pancreatic tissues.
We agree with the reviewer that the possibility to obtain ECM scaffolds from several organs and tissues represents a major development for tissue engineering and relative applications to regenerative medicine including the liver and the pancreas. However, the data presented in our manuscript are relative to an alternative application: engineering ECM scaffolds to obtain in vitro 3D cancer models (PDAC in this case) characterized by the ECM of the native microenvironment of the tumour. This is clearly introduced in the manuscript and we do not think there is any misunderstanding.
- Can the culturing of PDAC cells in 3D culture contribute to their better survival after a chemotherapeutic treatment versus a 2D culturing on plastic?
We thank the reviewer for raising this point. Indeed, it has been demonstrated by other groups that a 3D culture system is associated with an increase in chemoresistance. However, this effect was reported when employing a non-ECM-based 3D spheroid system (see as an example Longati P. et al) (1) where the resistance to Gemcitabine was lower than that observed in our study. In particular, using a higher concentration of Gemcitabine (1 uM), PANC1 cells were only 83% resistant compared to our study showing resistance of 93% ± 3% with a lower concentration (0.5uM).
Along these lines, the RNA-seq analysis presented in our manuscript suggests an active role of the ECM in the development of chemoresistance.
- In the “section 2.12. Treatment of 3D Bioengineered Scaffolds with Chemotherapeutics” experiment, the PDAC cells on plastic in 2D culture were used as a control. In my opinion, it is not entirely correct to compare the survival parameters in 2D and 3D culturing conditions. It is better to use the same culturing conditions for comparison, in this case, in 3D culture.
We agree with the reviewer that additional conditions involving non-ECM 3D cultures would have been instructive and this possibility was considered and discussed. However, we concluded that different non-ECM biomaterials would create different artificial non-well-characterised microenvironments affecting the RNA-seq results and not offering an appropriate term of comparison with the standard 2D culture systems. In addition, using 3D hydrogels such as Matrigel would not be appropriate as they include ECM from the basement membrane secreted by Engelbreth-Holm-Swarm (EHS) mouse sarcoma cells, hence affecting the biology of PDAC cells with signals that are not affine to the PDAC microenvironment. As much of this work is related to tissue-specific human ECM from pancreas as a primary site and liver as a metastatic site, it was that natural environment that we wanted to study, and 2D cells cultures were the comparative point as many of the studies on PDAC until today are largely based on that.
Furthermore, in previous studies we demonstrate tissue-specific ECM-related cell reactions, pointing to a strong and specific cell – ECM crosstalk (2–4)
- Has the data on the survival of the PDAC cells following the chemotherapeutic treatment been compared with the results of the chemoresistance of these cells on scaffolds? For example, after the gemcitabine treatment of PK1 cells cultured on a scaffold from liver, DNA damage (yH2A-) and apoptosis processes (ASP175-) were not detected. However, significant cell death was noted (lines 561-564) in this treatment variant.
Yes, many pathways were studied, unfortunately, we are unable to present and discuss them all within the limits of this manuscript. However, to specifically reply to this question, we checked our analysis on Gemcitabine-treated PK-1 cells on liver vs. pancreas scaffolds and indeed the RNAseq results reflect those of the viability, where the apoptosis pathway was significantly affected (p=9.663e-6) especially downstream genes within the pathway were almost exclusively upregulated in the liver in comparison to the pancreas scaffolds, even though h2afx (yH2A) and casp3 (ASP175) were both downregulated in the liver vs. the pancreas scaffolds.
- Based on the assumption made by the authors, there is a risk of developing oncological processes during the implantation of tissue-specific liver and pancreas scaffolds as part of tissue-engineered structures into damaged organs. Or does this assumption apply only to tissue tumors?
This is an excellent question. However, based on other work related to implanted scaffolds in mice and when other non-cancer cells were cultured on these scaffolds, it is reasonable to conclude that no there is no risk and this assumption only applied to tissue tumours. Along those lines, we have previously shown that implantation of healthy liver human ECM in immune-competent mice showed that there was no discernible immune reaction or rejection of the scaffold (5). Indeed, xenogenic-ECM derived from healthy tissues has been widely used in clinical trials for wound healing/tissue-repair strategy without reporting neoplastic transformation (6).
References:
- Longati P, Jia X, Eimer J, Wagman A, Witt MR, Rehnmark S, et al. 3D pancreatic carcinoma spheroids induce a matrix-rich, chemoresistant phenotype offering a better model for drug testing. BMC Cancer. 2013 Feb 27;13:95.
- Mazza G, Al-Akkad W, Telese A, Longato L, Urbani L, Robinson B, et al. Rapid production of human liver scaffolds for functional tissue engineering by high shear stress oscillation-decellularization. 2017 Jul 17;7(1):5534.
- Mazza G, Telese A, Al-Akkad W, Frenguelli L, Levi A, Marrali M, et al. Cirrhotic Human Liver Extracellular Matrix 3D Scaffolds Promote Smad-Dependent TGF-β1 Epithelial Mesenchymal Transition. Cells. 2019 Dec 28;9(1).
- Thanapirom K, Caon E, Papatheodoridi M, Frenguelli L, Al-Akkad W, Zhenzhen Z, et al. Optimization and validation of a novel three-dimensional co-culture system in decellularized human liver scaffold for the study of liver fibrosis and cancer. Cancers. 2021;13(19):4936.
- Mazza G, Rombouts K, Rennie Hall A, Urbani L, Vinh Luong T, Al-Akkad W, et al. Decellularized human liver as a natural 3D-scaffold for liver bioengineering and transplantation. Sci Rep. 2015 07 02/12/received 07/16/accepted;5:13079.
- Jiang Y, Li R, Han C, Huang L. Extracellular matrix grafts: From preparation to application (Review). Int J Mol Med. 2021 Feb;47(2):463–74.